# MM-Eureka: Exploring the Frontiers of Multimodal Reasoning with Rule-based Reinforcement Learning

## Abstract

DeepSeek-R1 and o1 have demonstrated powerful reasoning capabilities in the text domain through stable large-scale reinforcement learning. To enable broader applications, some works have attempted to transfer these capabilities to multimodal reasoning. However, these efforts have been limited by the limited difficulty of selected tasks and relatively small training scales, making it challenging to demonstrate strong multimodal reasoning abilities. To address this gap, we introduce the MMK12 dataset and MM-Eureka with 7B and 32B parameters. The former is a high-quality multimodal mathematics reasoning dataset featuring diverse knowledge domains with human-verified answers and solution processes. The latter is a multimodal model employing rule-based reinforcement learning on MMK12, utilizing online filtering and two-stage training strategy to enhance training stability. MM-Eureka demonstrates remarkable performance gains in multimodal mathematical reasoning, outperforming previous powerful models like InternVL2.5-78B or InternVL2.5-38B-MPO. In particular, MM-Eureka achieves competitive or superior performance compared to both open-source and closed-source models, and trails slightly behind o1 in multidisciplinary reasoning tasks. We open-source our complete pipeline to foster further research in this area. We release all our codes and data at `https://anonymous.4open.science/r/MMEUREKA-BDE0`.

## 1 Introduction

Large-scale reinforcement learning (RL) (Sutton et al., 1998) has demonstrated remarkable progress in improving the reasoning ability of Large Language Models (LLMs), particularly in the math and code domains (OpenAI, 2024; DeepSeek-AI et al., 2025). Recent research, such as o1 (OpenAI, 2024) and DeepSeek-R1 (DeepSeek-AI et al., 2025), shows that large-scale RL can achieve breakthrough improvements in complex reasoning tasks during post-training phases, sometimes even without supervised fine-tuning (SFT) (Radford et al., 2019). Despite great success in the text domain, many real-world reasoning tasks such as interpreting scientific diagrams and geometrical reasoning can only be effectively solved with the image input. However, transferring large-scale RL techniques that work well for LLMs to multimodal scenarios remains underexplored.

Recently, many works in the community have attempted to transfer the rule-based RL used in DeepSeek-R1 to multimodal scenarios. However, these works explore relatively small model sizes and fail to achieve stable training over extended periods like DeepSeek-R1. For instance, R1-V (Chen et al., 2025) demonstrates improvements only in simple counting tasks, which covers limited complexity, while the model size remains modest. LMM-R1 (Peng et al., 2025) achieves gains in accuracy reward for geometric reasoning; however, such success has not been verified in large-scale training with image-text data. Although Kimi k1.5 (Team et al., 2025a) has achieved competitive results in multimodal reasoning, it has not open-sourced its model or training data to the community. Therefore, how to achieve stable rule-based RL training in the multimodal reasoning domain remains an important and unsolved problem for the open-source community.

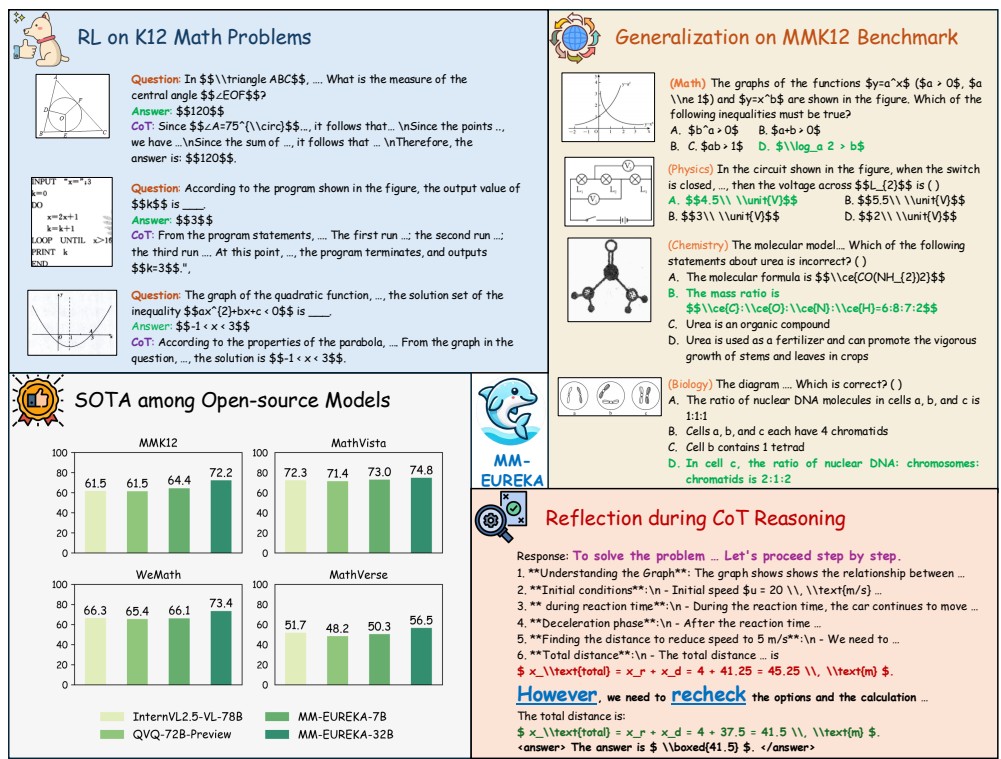

Figure 1: The overview of our proposed MMK12 and MM-Eureka. MMK12 training set has diverse multimodal mathematical questions with verified answer and process, while its evaluation set has multiple-choice questions for each discipline including math, physics, chemistry, and biology. MM-Eureka built on MMK12 has powerful performance in multimodal reasoning and it also exists aha-moment like DeepSeek-R1.

To bridge this gap, as shown in Figure 1, we first construct MMK12[1], a high-quality and diverse dataset for multimodal mathematical reasoning that covers K-12 level and olympiad-level problems. Building on MMK12, we develop two model variants—MM-Eureka-7B and MM-Eureka-32B. On the MMK12 evaluation set, MM-Eureka-7B reaches an accuracy of 64.5%, surpassing InternVL-2.5-78B (63.1%). MM-Eureka-32B further improves the score to 72.2%, outperforming both Qwen-VL-72B (70.5%) and the closed-source Gemini-Flash (71.0%), and is second only to o1 (73.9%). Across public benchmarks, MM-Eureka-32B achieves 73.4% on WeMath and 74.8% on MathVista, ranking first among all open-source models and exceeding most proprietary systems, including Claude-3.7-Sonnet. These results demonstrate that MM-Eureka delivers state-of-the-art performance in multidisciplinary K-12 and mathematical reasoning within its parameter range.

Specifically, despite growing interest in multimodal reasoning models, high-quality datasets for multimodal mathematical reasoning remain scarce. For example, Geo3k (Lu et al., 2021) and RCoT (Deng et al., 2024) focus only on geometry problems. Although MAVIS (Zhang et al., 2024b) comprises data involving geometry and functions, it consists of synthetic data, lacking diversity in images and questions. To this end, we construct MMK12, a K12-level multimodal mathematical reasoning dataset. The training set covers a wide range of domains, including functions, geometry, and equations, spanning from elementary through high school curricula. For evaluation, MMK12 includes four disciplines: mathematics, physics, chemistry, and biology. To facilitate testing, we collect 500 multimodal multiple-choice questions for each discipline and carefully verify that there is no overlap with the training set.

---

[1]K12 refers to the full span of primary and secondary education, and the K12 evaluation sets used in our work primarily contain questions from the secondary level.

Subsequently, to achieve stable rule-based RL training over extended periods, we utilize the online filter strategy from PRIME (Cui et al., 2025) that dynamically filters prompts with zero advantage during training, e.g., those answered either completely correctly or completely incorrect. For MM-Eureka-32B, we further introduce a two-stage training strategy: the first stage leverages MMK12 by RL without KL divergence to develop the model's general reasoning abilities while reducing training cost; the second stage fine-tunes the model on Geo3k with KL regularization to mitigate domain-specific deficiencies and enhance training stability. We use the GRPO (DeepSeek-AI et al., 2025) as our basic RL approach, which offers higher efficiency compared to the commonly used PPO (Schulman et al., 2017b). These strategies enable us to achieve stable, long-horizon RL training and consistent performance gains for both 7B and 32B model variants, while maintaining high training efficiency.

Through the journey of developing MM-Eureka, we have several findings. **First,** it is difficult for the model to acquire new knowledge through RL training. Instead, performance improvements come from increasing the probability that the model generates correct answers during inference. **Second,** we discover that simple rule-based RL exhibits outstanding generalization capabilities. Training solely on mathematical data leads to simultaneous improvements in physics, chemistry, biology, and other disciplines. **Third,** rule-based RL generalizes better than other post-training strategies such as SFT (Ouyang et al., 2022) and Chain-of-Thought (CoT) SFT (Guo et al., 2024) across various tasks.

Our goal is to share our implementation experiences and complete the open-source pipeline with the community, including data, code, and models. We believe this comprehensive open-source framework would help the community better explore the multimodal reasoning task. The main contributions are summarized as follows:

- We utilize an online filter strategy and introduce a two-stage training strategy to address the collapse issues encountered during RL training of large-scale VLMs, achieving stable rule-based RL training for large-scale VLMs.

- We present MM-Eureka-7B and MM-Eureka-32B. Extensive experimental results on various downstream tasks demonstrate that they are top performers among open-source models in the multimodal reasoning domain. For example, MM-Eureka-32B scores only 1.7 points below o1 on the multidisciplinary evaluation set of MMK12; MM-Eureka-7B achieves 73.0 on MathVista (Lu et al., 2024), surpassing InternVL2.5-78B (Chen et al., 2024).

- We open-source all our models, code, and collected high-quality multimodal mathematical reasoning data. Compared to existing open-source repositories, we support a wider range of RL algorithms and include much higher-quality data.

## 2 RELATED WORK

**Language reasoning model.** LLMs have demonstrated impressive performance across a wide range of tasks, yet more complex challenges require these models to exhibit human-like reasoning capabilities. As a result, enhancing the reasoning ability of LLMs has become a critical research focus. Reinforcement Learning from Human Feedback (RLHF), particularly Proximal Policy Optimization (PPO) (Schulman et al., 2017a), has shown promise in enabling LLMs to learn reasoning abilities effectively. However, the PPO training process is computationally intensive and complex, prompting the development of simplified alternatives such as Direct Preference Optimization (DPO) (Rafailov et al., 2023). While DPO alleviates some training difficulties, its reliance on offline data can limit model performance. To address these limitations, methods like Group Relative Policy Optimization (GRPO) (DeepSeek-AI et al., 2025), REINFORCE Leave-One-Out (RLOO) (Kool et al., 2019; Ahmadian et al., 2024), and Reinforce++ (Hu, 2025) have been introduced. Notably, DeepSeek-R1 (DeepSeek-AI et al., 2025) reveals that pure RL can encourage LLMs to actively engage in reasoning, including self-reflection and error correction. Despite these advancements, research on improving the reasoning capabilities of multimodal large models remains relatively scarce, highlighting an important direction for future exploration.

**Vision-language reasoning model.** Currently, the leading models in multimodal reasoning are closed-source systems such as GPT-4o (Hurst et al., 2024) and Kimi-VL (Team et al., 2025b). In contrast, the open-source community remains noticeably behind, still in the early stages of exploration. Recent concurrent efforts have begun to explore the use of RL to enhance the visual reasoning

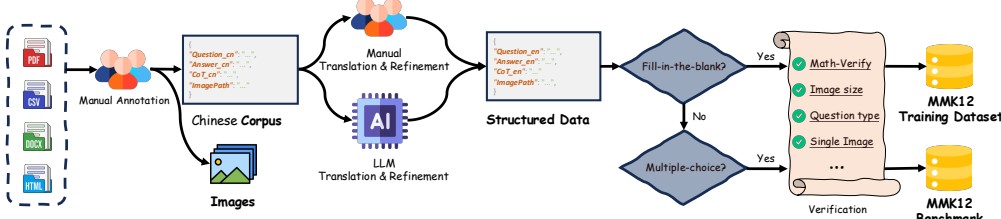

Figure 2: The construction overview of MMK12. We collect diverse K12-level multimodal math problems from multiple sources, convert them to standardized English using LLMs, and verify all content for accuracy. The resulting MMK12 dataset includes a training set of 15,616 samples and a test set with 500 multiple-choice questions each for math, physics, chemistry, and biology.

capabilities of vision-language reasoning models (VLMs), aiming to trigger an "Aha Moment" in visual reasoning. LMM-R1 (Peng et al., 2025) strengthens visual reasoning through a two-stage rule-based RL approach; however, its primary reasoning performance gains are derived from text-only datasets rather than genuinely multimodal datasets. R1-V (Chen et al., 2025) investigates rule-based RL within specific subdomains, such as geometric reasoning and object counting tasks, but falls short of addressing more complex reasoning challenges. Reason-RFT (Tan et al., 2025), on the other hand, relies on SFT with CoT reasoning activation data to achieve an effective cold start before the RL training phase. In this paper, our objective is to develop an effective, stable, and comprehensive open-source training pipeline for multimodal reasoning models, including datasets, code, and models. Our work aims to advance the growth and innovation of the open-source community.

## 3 MMK12: A MULTIMODAL MATHEMATIC K12-LEVEL DATASET

Table 1: Comparisons with other multimodal mathematical reasoning datasets. MMK12 comprises more diverse and high-quality questions, with guaranteed correct answers and solution processes. MCQ and FB refer to multiple-choice questions and fill-in-the-blank questions, respectively.

| | Scope | Type | Img. Source | QA Source | CoT Answer Source |
|---|---|---|---|---|---|
| MAVIS (Zhang et al., 2024b) | Geo & Func | MCQ & FB | Synthetic | Synthetic Engine | GPT-4o |
| Geo3k (Lu et al., 2021) | Geo | FB | Real World | Real World | None |
| RCoT (Deng et al., 2024) | Geo | MCQ & FB | Synthetic | Synthetic Engine | GPT-4o |
| MultiMath (Peng et al., 2024) | Diverse | MCQ & FB | Real World | GPT-4o | GPT-4o |
| **MMK12** | Diverse | FB | Real World | Real World | Real World |

As shown in Table 1, current multimodal mathematical reasoning datasets have limited scope and face challenges in ensuring answer correctness. For instance, while RCoT and MAVIS maintain answer accuracy through synthetic engine-generated QA pairs, this approach restricts problem diversity. Geo3k manually collected $3,000$ geometry problems with verified answers, but it focuses solely on geometry examples. Although MultiMath gathers problems from real-world scenarios to ensure diversity, its reference answers generated by GPT-4o cannot guarantee correctness.

To address these limitations, we introduce MMK12, a new dataset comprising over $15,000$ multimodal mathematical reasoning problems across a wide range of domains, including geometry, functions, and graphical reasoning. Each problem is accompanied by a standard reference answer and a detailed step-by-step solution to ensure both accuracy and interpretability.

In the following, we describe the construction and cleaning process of MMK12. As illustrated in Figure 2, we first collect a diverse set of multimodal mathematics problems and corresponding answers from a variety of Chinese mathematics textbooks and examination papers, covering grades from elementary to high school. These questions, answers, and CoT processes are subsequently translated and refined into English, with the help of LLM. To minimize false positives during training, we retain only fill-in-the-blank problems. Besides, we use Math-Verify[2] to parse the answer, ensuring data

---
[2]https://github.com/huggingface/Math-Verify

reliability for RL training. For the MMK12 evaluation set, we follow a similar construction procedure, but exclusively select multiple-choice questions to facilitate reliable and efficient evaluation.

In total, the MMK12 training dataset comprises $15,616$ multimodal fill-in-the-blank mathematics problems, including $455$ from elementary school, $9,776$ from middle school, and $5,385$ from high school. Each sample includes the question, an image, the final answer, and a CoT-formatted solution process. The MMK12 evaluation set has $2,000$ multimodal multiple-choice questions, including math, physics, chemistry, and biology. Each item includes a question, a corresponding image, and the correct answer choice. Some examples in MMK12 are shown in Figure 1. We introduce the data characteristics of MMK12 in more detail in Appendix B.

## 4 METHOD

### 4.1 BASIC SETTINGS

We use Qwen2.5-VL (Bai et al., 2023) with 7B and 32B parameters as our base models. Our reinforcement learning (RL) approach follows a similar design to DeepSeek-R1 (DeepSeek-AI et al., 2025), we also adopt the simple rule-based reward function rather than using outcome or process reward models, thereby alleviating reward hacking (Gao et al., 2022). Specifically, we employ rule-based rewards for output formatting ($r_{\text{format}} \in \{0, 0.5\}$) and accuracy ($r_{\text{accuracy}} \in \{0, 1\}$). More detail in Appendix C.1. More detail in Appendix C.1.

### 4.2 ADVANTAGE ESTIMATION AND POLICY UPDATE

GRPO (DeepSeek-AI et al., 2025) is a widely adopted RL algorithm that eliminates the need for training a complex critic model by leveraging intra-group relative performance to optimize the policy model. Specifically, for each query $\mathbf{x}$, the model generates a group of $G$ responses $\{\mathbf{y}^{(1)}, \mathbf{y}^{(2)}, \cdots, \mathbf{y}^{(G)}\}$. Subsequently, for each query with $G$ responses, GRPO computes the relative advantage of each response based on their rewards, which are determined by a reward model, as follows:

$$A^{(i)} = \frac{r^{(i)} - \text{mean}(\{r^{(j)}\}_{j=1}^{G})}{\text{std}(\{r^{(j)}\}_{j=1}^{G})}, \quad i = 1, \cdots, G.$$

Using the computed advantages, GRPO then optimizes the policy via the PPO-clip loss augmented with a directly imposed KL penalty term:

$$J_{\text{PPO}}(\theta) = -\mathbb{E}_{\mathbf{x} \sim \mathcal{D}, \{\mathbf{y}^{(i)}\}_{i=1}^{G} \sim \pi_\theta(\cdot|\mathbf{x})} \tag{1}$$

$$\left[ \frac{1}{G} \sum_{i=1}^{G} \frac{1}{|\mathbf{y}^{(i)}|} \sum_{t=1}^{|\mathbf{y}^{(i)}|} \left( \min\left( r^{i,t}(\theta)A^{(i)}, \text{clip}\left(r^{i,t}(\theta), 1-\epsilon, 1+\epsilon\right)A^{(i)} \right) - \beta D_{\text{KL}}^{i,t}(\pi_\theta, \pi_{\text{ref}}) \right) \right],$$

where

$$r^{i,t}(\theta) = \frac{\pi_\theta(y_t^{(i)}|\mathbf{x}, \mathbf{y}_{<t}^{(i)})}{\pi_{\theta_{\text{old}}}(y_t^{(i)}|\mathbf{x}, \mathbf{y}_{<t}^{(i)})} \text{ and } D_{\text{KL}}^{i,t}(\pi_\theta, \pi_{\text{ref}}) = \frac{\pi_{\text{ref}}(y_t^{(i)}|\mathbf{x}, \mathbf{y}_{<t}^{(i)})}{\pi_\theta(y_t^{(i)}|\mathbf{x}, \mathbf{y}_{<t}^{(i)})} - 1 - \log \frac{\pi_{\text{ref}}(y_t^{(i)}|\mathbf{x}, \mathbf{y}_{<t}^{(i)})}{\pi_\theta(y_t^{(i)}|\mathbf{x}, \mathbf{y}_{<t}^{(i)})}.$$

In addition, to mitigate loss spikes during training caused by excessively large policy ratios combined with negative advantages, we further constrain the policy ratio within $[0, c]$ beforehand, i.e., we replace $r^{i,t}(\theta)$ with $\text{clip}(r^{i,t}(\theta), 0, c)$ in Eq.(1). In practice, we set the default value of $c$ to 3.

### 4.3 ONLINE FILTERING

To further enhance training stability, we adopt the online prompt filtering strategy proposed in PRIME (Cui et al., 2025). To ensure sufficient gradient information throughout RL training, we filter out prompts with responses that are either completely correct or completely incorrect during training, as their corresponding advantages under GRPO are zero. See Appendix C.2 for the pseudocode.

We present an ablation study on the online filtering mechanism in Figure 3. Models trained with the online filter strategy maintain relatively stable trends in both the accuracy-based reward and response

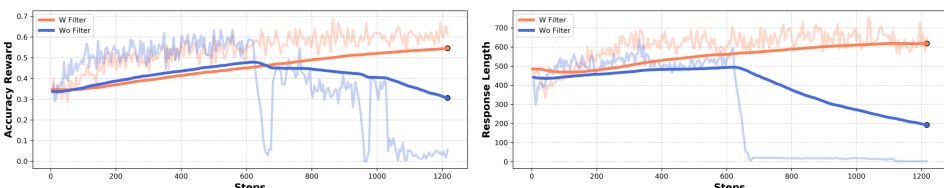

Figure 3: Comparison of the use of online filtering. Using online filter makes the RL training more stable in the long term, with consistently increasing accuracy reward and response length.

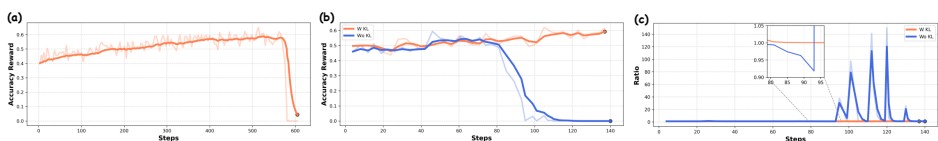

Figure 4: (a) MM-Eureka-32B tends to experience sudden training collapse during RL training, manifested as accuracy reward approaching zero. (b) During the second training phase, adding KL divergence leads to more stable training with steadily increasing accuracy rewards. (c) The stabilizing effect of KL divergence in the second phase may be attributed to the stabilization of the policy ratio, as ratio instability often leads to training collapse.

length throughout training. In contrast, models without the online filter show an initial improvement in accuracy, but soon experience a sharp decline, with accuracy eventually approaching zero and response lengths significantly shortening in the later stages. These findings suggest that the online filter plays an important role in stabilizing RL training and can help prevent model collapse during optimization.

## 4.4 TWO-STAGE TRAINING

During RL training of MM-Eureka-32B, we observe that maintaining training stability becomes increasingly challenging as model size scales up. Specifically, we find that models tend to experience sudden collapse during training, characterized by an abrupt drop of the accuracy reward to near-zero values, as shown in Figure 4 (a). After analysis, we find that these collapses are usually preceded by sharp increases in the policy ratio (indicating excessively large updates from old to new policies) like Figure 4 (c). Additionally, unlike the 7B model which shows comprehensive improvement when trained solely on K12 data, the 32B model trained only on MMK12 exhibits performance degradation in specific domains such as geometry.

To address these challenges, we propose a two-stage training strategy for large models. In the first stage, we apply the GRPO algorithm without KL regularization, using MMK12 data collected as detailed in Section 3 to broadly enhance the model's general reasoning capabilities. In the second stage, we incorporate a KL divergence into GRPO to constrain update magnitudes and improve training stability. Concurrently, we augment the training set with domain-specific Geo3k data to address identified performance gaps. As shown in Appendix E.2, this two-stage approach enables stable training and yields further performance gains across nearly all evaluated benchmarks.

## 5 EXPERIMENTS

We present our experimental setup in Section D.1, where we describe the training details. After that, we provide an overview of the selected baselines and evaluation benchmarks in Section D.2 and Section D.3. We demonstrate the superior effectiveness of MM-Eureka-7B and MM-Eureka-32B through extensive experiments in Section 5.1 below.

## 5.1 QUANTITATIVE RESULTS

In this section, we demonstrate the superiority of MM-Eureka through its performance on the benchmarks introduced in Section D.3. For the multimodal mathematical evaluation sets, we primarily use official results or results provided in VLMEvalKit (Duan et al., 2024) for baseline comparisons. When such results are unavailable, we conduct evaluations using the testing methodologies provided in VLMEvalKit within our vllm-based (Kwon et al., 2023) inference framework. Specifically, for o1, due to resource constraints, we randomly sample only 500 instances for testing. Since MMK12 consists entirely of multiple-choice questions, we adopt the same Chain-of-Thought prompt structure from WeMath (Qiao et al., 2024), which also uses only multiple-choice questions to evaluate models in our inference framework.

Table 2: Performance comparison across different multimodal mathematical benchmarks. **Bold** and underline indicate the best and the second-best performers among the open-source models, respectively. For some official results that are relatively different from our evaluation results, we mark our test results after '/'.

| Model | MathVista | MathVerse | MATH-Vision | OlypamidBench | WeMath |
|---|---|---|---|---|---|
| **Closed-Source Models** | | | | | |
| Claude3.7-Sonnet | 66.8 | 52.0 | 41.3 | 48.9 | 72.6 |
| GPT-4o | 63.8 | 50.2 | 30.4 | 35.0 | 68.8 |
| o1 | 73.9 | 57.0 | 60.3 | 68.0[3] | 98.7[3] |
| Gemini2-flash | 70.4 | 59.3 | 41.3 | 51.0 | 71.4 |
| **Open-Source General Models** | | | | | |
| InternVL2.5-8B | 64.4 | 39.5 | 19.7 | 12.3 | 53.5 |
| Qwen2.5-VL-7B | 68.2 | 47.9 | 25.4 | 20.2 | 62.1 |
| InternVL2.5-38B | 71.9 | 49.4 | 31.8 | 32.0 | 67.5 |
| Qwen2.5-VL-32B | 74.7/71.7 | 49.9 | **40.1** | 30.0 | 69.1 |
| InternVL2.5-78B | 72.3 | 51.7 | 32.2 | 31.1 | 66.3 |
| Qwen2.5-VL-72B | **74.8** | **57.6** | 38.1 | **40.4** | 72.4 |
| **Open-Source Reasoning Models** | | | | | |
| InternVL2.5-8B-MPO | 68.9 | 35.5 | 21.5 | 7.8 | 53.5 |
| InternVL2.5-38B-MPO | 73.8 | 46.5 | 32.3 | 25.6 | 66.2 |
| QVQ-72B-Preview | 71.4 | 48.2 | 35.9 | 33.2 | 65.4 |
| R1-Onevision-7B | 64.1 | 47.1 | 29.9/23.5[4] | 17.3 | 61.8 |
| OpenVLThinker-7B | 70.2 | 47.9 | 25.3 | 20.1 | 64.3 |
| **Ours** | | | | | |
| MM-Eureka-7B | 73.0 | 50.3 | 26.9 | 20.1 | 66.1 |
| MM-Eureka-32B | **74.8** | 56.5 | 34.4 | 35.9 | **73.4** |

**Mathematics**   In Table 2, we present a comprehensive summary of MM-Eurekas' performance across different multimodal mathematical reasoning tasks. It demonstrates that both MM-Eureka-7B and MM-Eureka-32B consistently outperform similar-sized open-source baselines across almost all tasks. In particular, MM-Eureka-7B achieves 73.0 on MathVista, surpassing InternVL2.5-78B by 0.7% and the reasoning-focused InternVL2.5-38B-MPO by 0.2%. On WeMath, it similarly approaches the performance of InternVL2.5-78B. MM-Eureka-32B exceeds all current open-source models of comparable size on all benchmarks except MATH-Vision, with average performance approaching Qwen2.5-VL-72B-Instruct. It even outperforms the closed-source model Claude3.7-Sonnet on WeMath. In summary, MM-Eureka establishes itself as the top performer among same-sized open-source models across almost all multimodal mathematical reasoning tasks, regardless of whether considering the 7B or 32B version.

---

[3]Results of o1 based on random sampling of 500 test instances due to resource constraints.

[4]Results of R1-Onevision in MATH-Vision is 23.5 from our evaluation, which is behind its official report as 29.9.

However, when compared to closed-source models, MM-Eureka-32B still shows considerable performance gaps. While it outperforms o1 on MathVista, it lags significantly behind closed-source multimodal reasoning models like o1 on more challenging benchmarks such as MATH-Vision or OlympiadBench.

Table 3: Performance comparison across different disciplines in MMK12. Performance comparison across different multimodal mathematical benchmarks. **Bold** and underline indicate the best and the second-best performers among the open-source models, respectively.

| Model | Mathematics | Physics | Chemistry | Biology | Avg. |
|---|---|---|---|---|---|
| **Closed-Source Models** | | | | | |
| Claude3.7-Sonnet | 57.4 | 53.4 | 55.4 | 55.0 | 55.3 |
| GPT-4o | 55.8 | 41.2 | 47.0 | 55.4 | 49.9 |
| o1 | 81.6 | 68.8 | 71.4 | 74.0 | 73.9 |
| Gemini2-flash | 76.8 | 53.6 | 64.6 | 66.0 | 65.2 |
| **Open-Source General Models** | | | | | |
| InternVL2.5-8B | 46.8 | 35.0 | 50.0 | 50.8 | 45.6 |
| Qwen2.5-VL-7B | 58.4 | 45.4 | 56.4 | 54.0 | 53.6 |
| InternVL2.5-38B | 61.6 | 49.8 | 60.4 | 60.0 | 58.0 |
| Qwen2.5-VL-32B | 71.6 | 59.4 | 69.6 | 66.6 | 66.8 |
| InternVL2.5-78B | 59.8 | 53.2 | 68.0 | 65.2 | 61.6 |
| Qwen2.5-VL-72B | **75.6** | **64.8** | 69.6 | 72.0 | 70.5 |
| **Open-Source Reasoning Models** | | | | | |
| InternVL2.5-8B-MPO | 26.6 | 25.0 | 42.4 | 44.0 | 34.5 |
| InternVL2.5-38B-MPO | 41.4 | 42.8 | 55.8 | 53.2 | 48.3 |
| QVQ-72B-Preview | 61.4 | 57.4 | 62.6 | 64.4 | 61.5 |
| R1-Onevision | 44.8 | 33.8 | 39.8 | 40.8 | 39.8 |
| OpenVLThinker-7B | 63.0 | 53.8 | 60.6 | 65.0 | 60.6 |
| **Ours** | | | | | |
| MM-Eureka-7B | 71.2 | 56.2 | 65.2 | 65.2 | 64.5 |
| MM-Eureka-32B | 74.6 | 62.0 | **75.4** | **76.8** | **72.2** |

**MMK12** Beyond validating our model's superiority on widely-used multimodal mathematical reasoning benchmarks like MathVista, it is necessary to test its capabilities and generalization across multidisciplinary reasoning domains using questions absent from the training set (e.g., physics, chemistry, and biology). For this purpose, we employ the MMK12 dataset constructed in Section 3, which can effectively measure models' multimodal reasoning capabilities across multiple disciplines.

As shown in Table 3, MM-Eureka-32B demonstrates multidisciplinary capabilities only marginally behind o1 by 1.7%, while outperforming larger-scale models such as Qwen2.5-VL-72B and Gemini2-Flash-Thinking. MM-Eureka-7B also surpasses several similarly-sized multimodal reasoning models, including OpenVLThinker-7B, with overall performance exceeding InternVL2.5-78B and only slightly behind Qwen2.5-VL-32B. Additionally, we observe several interesting findings: 1) Despite being trained exclusively on fill-in-the-blank questions, our models maintain strong instruction-following abilities for multiple-choice questions with improved performance. 2) Even with training solely on mathematics problems, the models exhibit enhanced capabilities in physics, chemistry, and biology. Specifically, MM-Eureka-7B shows improvements of 9.8% and 11.2% in chemistry and biology.

## 5.2 QUALITATIVE RESULTS

Appendix F presents representative examples comparing MM-Eureka-32B with its base model, Qwen2.5-VL-32B-Instruct, across four subjects: mathematics, physics, chemistry, and biology. These qualitative cases highlight the significant gains in reasoning ability achieved by MM-Eureka-32B after reinforcement learning. In particular, our model demonstrates a stronger capacity to apply known concepts and carry out multi-step deductions to arrive at correct solutions. By contrast, the

base model often exhibits only superficial understanding—recalling relevant facts but failing to integrate them effectively in problem-solving scenarios. Additional analysis and examples can be found in Appendix F.

## 6 DISCUSSION

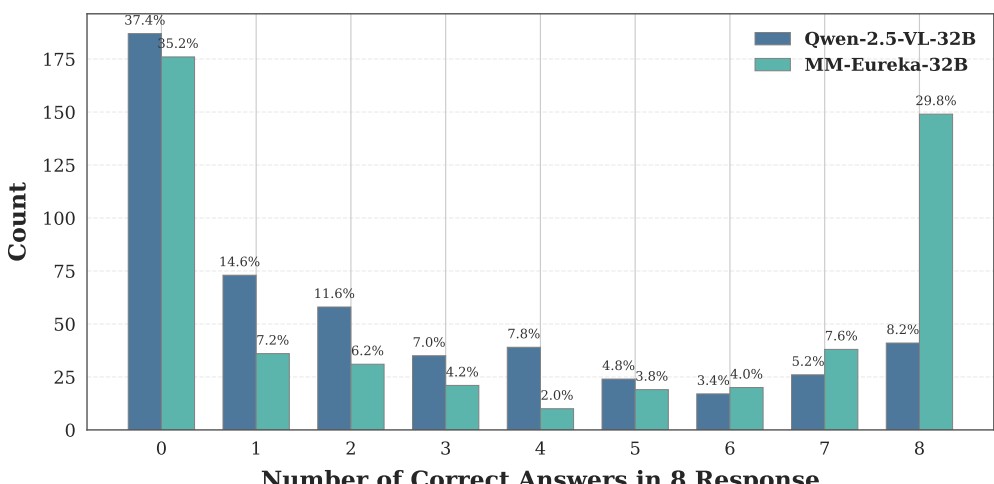

Figure 5: Distribution of correct answers across 8 responses from Qwen2.5-VL-32B and MM-Eureka-32B on *Mathematics*.

**Are knowledge and reasoning decoupled?** Figure 5 presents the distribution of correct answers across 8 responses from the Qwen2.5-VL-32B and MM-Eureka-32B models on the *Mathematics* dataset. For problems that received at least one correct response initially, MM-Eureka-32B demonstrates a clear improvement in accuracy. We attribute this gain to enhanced reasoning capabilities developed through reinforcement learning, which allow the model to better leverage acquired knowledge and refine its answers. In contrast, for problems that consistently received zero correct responses, the accuracy remains nearly unchanged. This suggests that improved reasoning alone is insufficient when the model lacks the necessary knowledge foundation, highlighting a bottleneck where further gains may require explicit knowledge acquisition rather than better inference.

From the responses of Qwen2.5-VL and MM-Eureka in Figure 6 in the Appendix, we observe that although Qwen2.5-VL possesses the necessary knowledge to answer the question, it still struggles to provide the correct answer. In contrast, MM-Eureka is able to better reason with the available knowledge and derive the correct answer. Table 8 further illustrates that, despite MM-Eureka's training set being focused only on mathematics, it also demonstrates a notable improvement in answering questions from other domains such as physics, chemistry, and biology. Our experimental results to some extent support the view that knowledge and reasoning can be decoupled, and further indicate that the two components may be learned independently during training.

**More discussion.** We provide a more detailed discussion in Appendix E, including a comparison between RL and SFT, as well as the effectiveness of the two-stage training approach.

## 7 CONCLUSION

In this work, we aim to develop a strong model for multimodal reasoning. To this end, we introduce MMK12, a high-quality mathematical dataset designed to support complex multimodal reasoning tasks. In addition, we employ an online filtering mechanism and propose a two-stage training strategy to enhance the stability and efficiency of rule-based reinforcement learning. Based on these contributions, we present MM-Eureka-7B and MM-Eureka-32B, which achieve state-of-the-art performance among models of comparable size on multimodal reasoning benchmarks.

## 8  ETHICS STATEMENT

This work adheres to the ICLR Code of Ethics. No human subjects or animal experimentation was involved in this study. The MMK12 dataset used complies with relevant usage guidelines, ensuring no privacy violations. We carefully avoided biases and discriminatory outcomes when utilizing the QwenVL model. No personally identifiable information was used, and no experiments were conducted that could raise privacy or security concerns. We are committed to maintaining transparency and integrity throughout the research process.

## 9  REPRODUCIBILITY STATEMENT

We have made every effort to ensure the reproducibility of our results. All code and datasets are publicly available in an anonymous repository (`https://anonymous.4open.science/r/ MMEUREKA-BDE0`) to facilitate reproduction and verification. This paper provides detailed descriptions of experimental settings, including training procedures, model configurations, and hardware specifications. We also provide complete descriptions of experimental runs in the anonymous repository to help others reproduce our experiments. Additionally, the MMK12 training dataset used is fully open, and the QwenVL training model employed is open-source, ensuring consistency and reproducibility of evaluation results. We believe these measures enable other researchers to reproduce our work and further advance the field.

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

## A   THE USE OF LARGE LANGUAGE MODELS (LLMs)

LLMs were utilized in the preparation of this manuscript to refine language quality and address grammatical issues. The specific prompt applied was: "Please identify and rectify grammatical mistakes in the provided text, while improving its scholarly presentation and clarity."

## B   MMK12

### B.1   OVERVIEW

MMK12 encompasses mathematical problems across various knowledge domains, including geometry, functions, spatial reasoning, and more. Some important categories are introduced as follows:

- Function Reasoning: This task requires models to understand function concepts, analyze function graphs and expressions, and apply function properties to solve problems. This type of reasoning develops the model's ability to comprehend abstract mathematical concepts, fostering its capability to identify function characteristics, determine critical points, and analyze function behavior.

- Geometric Reasoning: This task involves applying spatial relationships, geometric theorems, and properties of shapes. Through geometric reasoning training, models enhance their spatial visualization, logical deduction, and formalization abilities for geometry problems, enabling them to solve complex problems in both plane and solid geometry.

- Pattern Reasoning: This type of task focuses on understanding flow diagrams and recognizing patterns in visual sequences. Models need to discover patterns, predict rule-based changes, or understand logical relationships in visual content. This task examines the model's pattern recognition abilities, inductive reasoning skills, and visual logical thinking.

Benefiting from a standardized data construction process with substantial human involvement and verification, we highlight the unique advantages of MMK12 compared to other multimodal reasoning datasets through the following aspects:

- Real-world data with guaranteed correctness: Unlike MathV360k (contains synthetic data limiting diversity) and MultiMath (LLM-generated answers without verification), MMK12 is the only multimodal math dataset that combines rich categories, real-world sampling, and verified answer correctness.

- Reliable COT support for both RL and SFT: While existing datasets like MAVIS use LLM-generated COT (unreliable) or RCOT use template-based rewriting (limited diversity), MMK12's COT annotations come from manually verified real-world reference solutions, ensuring high reliability.

- Comprehensive mathematical coverage: Unlike narrow datasets like Geo3k (geometry only, few samples), MMK12 spans five major categories as shown in Table 1 below, providing rich variety.

- Universal model compatibility: While concurrent works like R1-Onevision and OpenVLThinker have proposed a dataset but only validate on QwenVL, we demonstrate MMK12's effectiveness across multiple multimodal models (QwenVL, InternVL) in Table 2 below, proving its universality.

### B.2   DETAILS

**Statistics**   We categorize different types of tasks in MMK12 in Table 4 and provide statistical information on their proportions.

Additionally, we supplement Table 5 with the final model performance when removing any specific task type. As shown, each task contributes positively to the overall performance, demonstrating the importance of MMK12's diverse task composition.

**Qualitative analysis**   Finally, we provide specific qualitative comparisons through examples to highlight the advantages of MMK12 in Table 6. Due to space limitations, we randomly selected some examples from MultiMath (Peng et al., 2024), which also samples from the real world. It can be seen that MultiMath directly uses LLM-generated answers that don't even provide the final results. In

Table 4: Distribution of task types in the MMK12 dataset.

| Type | Count | Percentage |
|------|-------|------------|
| Functions | 1586 | 10.2% |
| Geometry | 9819 | 62.9% |
| Statistics and Probability | 1734 | 11.1% |
| Algorithms and Flowcharts | 1595 | 10.2% |
| Other | 882 | 5.6% |

Table 5: Ablation study showing model performance across different benchmarks when removing specific task types from MMK12.

| Model | MathVista | MathVerse | MATH-Vision | OlympiadBench | WeMath | MMK12 | Overall |
|-------|-----------|-----------|-------------|---------------|--------|-------|---------|
| Qwen2.5-VL-7B | 68.2 | 47.9 | 25.4 | 20.2 | 62.1 | 53.6 | 46.2 |
| MMK12 | 73.0 | 50.3 | 26.9 | 20.1 | 66.1 | 64.5 | 50.2 |
| - Algorithms | 69.8 | 52.7 | 26.7 | 21.4 | 65.9 | 61.5 | 49.6 |
| - Functions | 68.0 | 49.5 | 19.4 | 22.3 | 65.5 | 63.0 | 47.9 |
| - Geometry | 70.6 | 49.1 | 22.2 | 22.6 | 68.0 | 59.4 | 48.7 |
| - Statistics | 71.2 | 50.2 | 24.4 | 22.2 | 66.0 | 61.9 | 49.3 |
| - Other | 72.5 | 50.1 | 19.1 | 20.2 | 63.0 | 60.1 | 47.5 |

contrast, MMK12 has clearer and more reliable problems and solution processes. We will add more comparisons between MMK12 and other datasets, as well as visualizations!

Table 6: Some examples from MMK12 and MultiMath. Due to space limitations, some texts have been abbreviated with middle sections omitted.

| Dataset | Problem | Answer |
|---------|---------|--------|
| MultiMath | Here is a math problem. The problem statement is as follows: $\angle ABD = \angle CBD$, $DF \parallel AB$, $DE \parallel BC$, the size relationship of $\angle 1$ and $\angle 2$ is___. | The image contains a triangle labeled ABC. Point D is on side AC, with line segment BD extending from point B and intersecting AC at point D..., as well as two smaller split angles which are near point D and are labeled "1" and "2" (indicating $\angle 1$ and $\angle 2$). |
| MultiMath | Here is a math problem. The problem statement is as follows: if input $n, x$ is 3, 3, and the output $v$ is (___). | This image depicts a flowchart for the Horner's Rule algorithm. At the top, there's a "Start" symbol...If $i$ equals 0, the loop ends, leading to an "Output $v$" statement, representing the output of the value of $v$, followed by an "End" symbol. |
| MMK12 | As shown in the figure, point O is a point on line AD ray, OC and OE are the bisectors of $\angle AOB$ and $\angle BOD$ respectively. If $\angle AOC = 25$, then $\angle BOE = $ ___. | Since $\angle AOB = 2\angle AOC = 50$, and $\angle AOB + \angle BOD = 180$, therefore $\angle BOD = 130$, so $\angle BOE = \frac{1}{2}\angle BOD = 65$ |

# C  METHOD

## C.1  BASIC SETTINGS

The training is conducted using GRPO as the base RL algorithm. To support multimodal inputs, we develop a custom RL framework built on OpenRLHF (Hu et al., 2024). Following DeepSeek-R1, we also adopt the simple rule-based reward function rather than using outcome or process reward models, thereby alleviating reward hacking (Gao et al., 2022). Specifically, we use two types of rewards: accuracy reward and format reward. The former uses Math-Verify to extract the answer from model responses and compare it with the reference one, returning 1 or 0 based on correctness; the latter checks whether the response follows the specified format (`<think>...</think><answer>...</answer>`), returning 0.5 or 0 based on compliance. We

find that this simple and sparse reward is sufficient to significantly improve the model's multimodal reasoning ability.

### C.2 Pseudocode of Online Filter

---

**Algorithm 1** Online Filter

---

**Input**: Prompt dataset $\mathcal{D}$; initial policy $\pi_\theta$; reward model $\boldsymbol{R}$; buffer size $N_B$; hyperparameters $K_1, K_2, G, \epsilon_{\text{acc}}^{\text{lower}}, \epsilon_{\text{acc}}^{\text{upper}}$

**Output**: $\pi_\theta$

0: Buffer $\mathcal{B} \leftarrow \{\}$
0: **for** iteration $= 1, \ldots, K_1$ **do**
0:     Sample a batch of prompts $\mathcal{Q} \sim \mathcal{D}$
0:     **for** each prompt $\mathbf{x} \in \mathcal{Q}$ **do**
0:         Generate $G$ responses $\{\mathbf{y}^{(i)}\}_{i=1}^G \sim \pi_\theta(\cdot|\mathbf{x})$
0:         Compute the reward $r^{(i)} = \boldsymbol{R}(\mathbf{x}, \mathbf{y}^{(i)})$ for $i = 1, \ldots, G$
0:         Compute the accuracy $\mathcal{C}_\mathbf{x} = |\{\mathbf{y}^{(i)}|r^{(i)} = 1\}|$
0:         **if** $\epsilon_{\text{acc}}^{\text{lower}} \leq \mathcal{C}_\mathbf{x} \leq \epsilon_{\text{acc}}^{\text{upper}}$ **then**
0:             Add the sample to the buffer $\mathcal{B} \leftarrow \mathcal{B} \cup \{(\mathbf{x}, \mathbf{y}^{(i)}, r^{(i)})\}_{i=1}^G$
0:         **end if**
0:     **end for**
0:     **if** $|\mathcal{B}| \geq N_B$ **then**
0:         **for** Epoch $= 1, \ldots, K_2$ **do**
0:             Update policy $\pi_\theta$ on $\mathcal{B}[0{:}N_B]$ by any RL algorithm
0:         **end for**
0:         Buffer $\mathcal{B} \leftarrow \{\}$
0:     **end if**
0: **end for**=0

---

## D Experiment

### D.1 Experiments Setup

We design our prompt template following the format used in DeepSeek-R1, wherein the system prompt explicitly specifies the required output structure, including the use of `<answer>` tags to separate the reasoning process from the final answer. Detailed prompt configurations are provided in Table 7.

For training hyperparameters, both the rollout batch size and training batch size are set to $128$, with $8$ rollouts generated per sample. Sampling is conducted with a temperature of $1.0$ to encourage response diversity, and optimization is performed using a learning rate of $1 \times 10^{-6}$. Besides, for MM-Eureka-32B, we adopt a two-stage training scheme. In Stage 1, KL divergence is disabled to promote exploration and policy flexibility during the early phase. In Stage 2, a small KL penalty $1 \times 10^{-3}$ is introduced to stabilize training.

All training is performed on H100 GPU. For 7B training, 24 H100s are needed to run for about 20 hours. For 32B training, 48 H100s are needed to run for about 48 hours.

### D.2 Baselines

To comprehensively evaluate the effectiveness of MM-Eureka, we compare it against a diverse set of baselines, including both closed-source and open-source systems.

**Closed-Source Models.** We include several leading proprietary models in our comparison. GPT-4o (Hurst et al., 2024), Claude3.7-Sonnet (Anthropic, 2024), and Gemini2-flash (Team et al., 2023) are general multimodal models without reasoning optimization, while o1 (OpenAI, 2024) is explicitly designed to enhance reasoning capabilities. These models serve as strong references for the current state-of-the-art.

Table 7: Prompt setting for MM-Eureka.

| |
| --- |
| **SYSTEM:** Solve the question. The user asks a question, and you solve it. You first think about the reasoning process in the mind and then provide the user with the answer. The answer is in latex format and wrapped in $...$. The final answer must be wrapped using the \boxed{} command. Th answer should be enclosed within <answer></answer>tags, i.e., Since $1+1=2$, so the answer is $2$. <answer>The answer is $\boxed{2}$ </answer>, which means the final answer assistant's output should start with <answer>and end with </answer>.
**USER:** <image>{{question}} |

**Open-Source General Models.**   This group comprises high-capacity vision-language models trained for general purposes, including Qwen2.5-VL (Bai et al., 2023) and InternVL2.5 (Chen et al., 2024) across various model sizes (7B to 78B). These models are primarily pretrained or instruction-tuned on large-scale image-text datasets. They offer a baseline for evaluating the impact of reasoning-specific post-training.

**Open-Source Reasoning Models.**   We further include open-source models explicitly fine-tuned for reasoning, such as InternVL2.5-MPO variants (Wang et al., 2024b), QVQ-72B-Preview (Team, 2024), R1-Onevision (Yang et al., 2025) , and OpenVLThinker (Deng et al., 2025). These models employ various strategies, including SFT (Ouyang et al., 2022), DPO (Rafailov et al., 2023), and rule-based reinforcement learning. They represent the most competitive open-source efforts in multimodal reasoning.

### D.3 BENCHMARKS

We conduct evaluations on multiple benchmark datasets, including MathVista(testmini) (Lu et al., 2024), MathVerse(testmini) (Zhang et al., 2024a), MATH-Vision(test) (Wang et al., 2024a), OlympiadBench(EN-OE split) (He et al., 2024) and WeMath (Qiao et al., 2024). MathVista is one of the most widely used multimodal mathematical benchmarks, offering a diverse set of problems that span general visual question answering, figure question answering, logic, algebra, and geometry. MathVerse, on the other hand, focuses specifically on the model's ability to comprehend images, with tasks categorized into areas such as algebra and geometry. MATH-Vision takes this a step further by emphasizing more abstract visual understanding, testing the model's capacity for recognizing and reasoning beyond conventional mathematical contexts. OlympiadBench presents graduate-level mathematical competition problems, from which we select the English questions for evaluation. WeMath complements these by providing fine-grained diagnostic insights into model reasoning behavior, using a large-scale, hierarchically annotated problem set to assess knowledge mastery and generalization.

Beyond these established benchmarks, both our model and the baselines are further evaluated on the proposed MMK12 Benchmark, as introduced in Section 3. Unlike OlympiadBench, which targets advanced graduate-level mathematical reasoning, MMK12 focuses on assessing the model's ability to solve fundamental multimodal multidisciplinary problems commonly encountered in K12 education. MMK12 enables a comprehensive evaluation of a model's ability across disciplines beyond mathematics.

During evaluation, we adopt greedy decoding with a temperature of $0$, ensuring deterministic outputs. We do not use beam search, and both top-$p$ and top-$k$ sampling are disabled.

## E   DISCUSSION

### E.1   RL GENERALIZES BETTER THAN SFT

We maintain consistent settings with our RL training to compare different post-training strategies including SFT and CoT SFT. Using the ms-swift framework (Zhao et al., 2024), we conduct both SFT and CoT SFT training for 10 epochs with identical data. As shown in Table 8, results demonstrate that RL exhibits superior generalization compared to SFT or CoT SFT approaches, particularly with more

significant improvements on OOD test sets such as Physics, Chemistry, and Biology. Furthermore, RL enhances the model's reasoning capabilities more effectively—while SFT and CoT SFT fail to substantially improve performance on mathematics and physics problems, RL training increases the model's scores in mathematics and physics by 12.8 and 10.8 points respectively.

Table 8: Performance comparison of different training methods on MMK12. In terms of both enhancing mathematical capabilities and generalizing to other disciplines, RL significantly outperforms SFT or CoT SFT.

| Model | Mathematics | Physics | Chemistry | Biology | Avg. |
|---|---|---|---|---|---|
| Qwen2.5-VL-7B | 58.4 | 45.4 | 56.4 | 54.0 | 53.5 |
| + SFT | 56.6 | 50.0 | 63.2 | 61.2 | 57.7 |
| + CoT SFT | 59.2 | 46.0 | 62.2 | 61.2 | 57.1 |
| + RL | **71.2** | **56.2** | **65.2** | **65.0** | **64.5** |

### E.2 TWO-STAGE TRAINING

In Section 4.4, we introduce a two-stage RL training strategy. Stage 1 employs K12 data to enhance general knowledge reasoning capabilities, while stage 2 utilizes Geo3k data to address specific performance gaps. Additionally, we omit the KL divergence in stage one to accelerate training and then incorporate it in stage two to maintain training stability.

As shown in Figure 4 (a), continuous training without KL divergence often leads to sudden model collapse. We identify this phenomenon results from ratio instability (excessive updates in the new policy). As shown in Figures 4 (b) and (c), which present training metrics with and without the use of KL divergence in the second stage, a sharp fluctuation in the ratio is observed when the model begins to collapse. To mitigate this issue of ratio instability, we introduce a KL divergence term during the second stage of training, which helps to stabilize the learning dynamics.

Furthermore, Table 9 demonstrates that after the first training stage, despite overall performance improvements, capabilities in certain specific domains actually decline, as evidenced by MathVista's geometry problem solving(GPS) score dropping from 74.0 to 56.7. To address this, we implement second-stage training using Geo3k, which yields further improvements across multiple mathematical benchmarks.

Table 9: Performance comparison across multimodal mathematical benchmarks of the two stage training strategy. The second phase of training enhances various capabilities of the model following the first phase, with particularly significant improvements in geometric reasoning abilities.

| Model | Data Scale | MathVista | MathVista(GPS) | MathVerse | MATH-Vision | WeMath |
|---|---|---|---|---|---|---|
| Stage1 | 15k | 72.0 | 56.7 | 56.4 | 33.6 | **73.5** |
| Stage2 | 2k | **74.8** | **70.2** | **56.5** | **34.4** | 73.4 |

We also provide the score changes for all categories in MathVista after the second stage training in Table 10. We can see that almost all subcategories show improvements after the two-stage training. The slight decreases in figure QA and statistical reasoning are attributed to the large domain gap between these tasks and geometry problems.

Table 10: Performance improvements ($\Delta$) across different skill categories in MathVista after Stage 2 training

| Skills | Sci | TQA | Num | Common | Arith | VQA | Geo | Algebra | GPS | MWP | Logic | FQA | Stat |
|---|---|---|---|---|---|---|---|---|---|---|---|---|---|
| $\Delta$ | +2.8 | +1.7 | +0.6 | +1.4 | +0.6 | +0.6 | +11.0 | +8.6 | +13.5 | +2.7 | +0.0 | -2.2 | -0.7 |

## F QUALITATIVE ANALYSIS

Figures 6, 7, 8, and 9 present examples from mathematics, physics, chemistry, and biology, respectively. For instance, in the physics example (Figure 7), the question involves analyzing the

instantaneous contact force between two objects placed on a vertical spring. While both models recognize key physical quantities such as mass, weight, and spring force, only MM-Eureka-32B correctly applies Newton's second law to derive the acceleration of the system and compute the contact force. It identifies that the net external force due to the added mass leads to a downward acceleration, and accurately calculates the resulting contact force between object A and B as $24N$. In contrast, the base model Qwen2.5-VL-32B-Instruct incorrectly concludes that the contact force equals the full weight of object B (i.e., $40N$), failing to consider the transient acceleration of the system and misapplying Newtonian mechanics.

These examples highlight MM-Eureka's improved ability to perform reasoning and its potential to generalize reasoning patterns across disciplines. Further cases in mathematics, chemistry, and biology show similar trends, where our model more effectively breaks down problems into intermediate steps and synthesizes knowledge to reach accurate conclusions.

## G LIMITATIONS

Although this work achieved stable rule-based RL through a novel training strategy and trained an outstanding model in the field of multimodal reasoning, it still has some limitations, such as not exploring larger model structures (such as 72B) and lacking multi-language reasoning capabilities.

**Question:**
As shown in the figure, in the straight square column $ABCD - A_1B_1C_1D_1$, $\angle ADC = 90°$, and $AA_1 = AD = DC = 1$, $M \in$ plane $ABCD$, when $D_1M \perp$ plane $A_1C_1D$, $D_1M = $___.

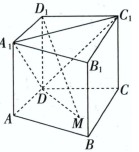

**Answer:** $\sqrt{3}$

---

🌀 **Qwen2.5-VL-32B-Instruct**

To solve the problem, we need to determine the length of $D_1M$, where $M$ is a point in the plane $ABCD$ such that $D_1M \perp$ plane $A_1C_1D$.

...

**2. Coordinates of Points:** Place the prism in a 3D Cartesian coordinate system:

$$A = (0,0,0), \quad B = (1,0,0), \quad C = (1,1,0), \quad D = (0,1,0)$$

$$A_1 = (0,0,1), \quad B_1 = (1,0,1), \quad C_1 = (1,1,1), \quad D_1 = (0,1,1)$$

Let $M = (x, y, 0)$, where $M$ is in the plane $ABCD$.

**3. Vector Representation:** $\overrightarrow{D_1M} = (x - 0, y - 1, 0 - 1) = (x, y - 1, -1)$

**4. Condition for Perpendicularity:** For $D_1M \perp$ plane $A_1C_1D$, the vector $\overrightarrow{D_1M}$ must be perpendicular to the normal vector of the plane $A_1C_1D$.

First, find two non-parallel vectors in the plane $A_1C_1D$:

...

Therefore, the length of $D_1M$ is $\boxed{\dfrac{\sqrt{6}}{2}}$.

---

🐢 **MM-EUREKA-32B**

To solve the problem, we need to determine the length of $D_1M$ when $D_1M \perp$ plane $A_1C_1D$. Let's break the problem into steps:

...

**2. Determination of the plane $A_1C_1D$:**

- The points $A_1 = (1, 0, 1)$, $C_1 = (0, 1, 1)$, and $D = (0, 0, 0)$ lie in the plane $A_1C_1D$.
- To find the normal vector to the plane $A_1C_1D$, we use the cross product of two vectors in the plane:

$$\overrightarrow{A_1D} = (1 - 0, 0 - 0, 1 - 0) = (1, 0, 1), \quad \overrightarrow{C_1D} = (0 - 0, 1 - 0, 1 - 0) = (0, 1, 1).$$

- The normal vector $\mathbf{n}$ to the plane is given by the cross product:

$$\mathbf{n} = \overrightarrow{A_1D} \times \overrightarrow{C_1D} = \begin{vmatrix} \mathbf{i} & \mathbf{j} & \mathbf{k} \\ 1 & 0 & 1 \\ 0 & 1 & 1 \end{vmatrix} = \mathbf{i}(0 \cdot 1 - 1 \cdot 1) - \mathbf{j}(1 \cdot 1 - 1 \cdot 0) + \mathbf{k}(1 \cdot 1 - 0 \cdot 0) = (-1, -1, 1).$$

- Thus, the normal vector to the plane $A_1C_1D$ is $\mathbf{n} = (-1, -1, 1)$.

**3. Condition for $D_1M \perp$ plane $A_1C_1D$:**

- Since $D_1M \perp$ plane $A_1C_1D$, the vector $\overrightarrow{D_1M}$ must be parallel to the normal vector $\mathbf{n} = (-1, -1, 1)$.
- Let $M = (x, y, 0)$ be a point in the plane $ABCD$ (since $M \in$ plane $ABCD$, the $z$-coordinate of $M$ is 0).
- The vector $\overrightarrow{D_1M} = (x - 0, y - 0, 0 - 1) = (x, y, -1)$.

...

**4. Calculation of $D_1M$:**

- The vector $\overrightarrow{D_1M} = (1 - 0, 1 - 0, 0 - 1) = (1, 1, -1)$.
- The length of $D_1M$ is given by the magnitude of $\overrightarrow{D_1M}$: $|\overrightarrow{D_1M}| = \sqrt{1^2 + 1^2 + (-1)^2} = \sqrt{1 + 1 + 1} = \sqrt{3}$.

<answer>The answer is $\boxed{\sqrt{3}}$. </answer>

---

Figure 6: Example from mathematics. Red highlights Qwen2.5-VL-32B-Instruct's reasoning errors. Green shows MM-Eureka-32B's correct reasoning.

**Question:**
As shown in the figure, a lightweight spring is placed vertically on a horizontal surface with a mass of $6kg$ object $A$ resting on its upper end, in a stationary state. Now, a mass of $4kg$ object $B$ is gently placed on top of object $A$, and the spring remains within its elastic limit. Taking $g$ as $10m/s^2$, at the moment $B$ is placed, the magnitude of the force between $A$ and $B$ is ( ) A.

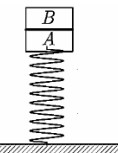

$8N$ B. $18N$ C. $24N$ D. $40N$

**Answer:** $C$

---

🦜 **Qwen2.5-VL-32B-Instruct**

To solve this problem, we need to determine the force between objects $A$ and $B$ at the moment $B$ is placed on top of $A$. Let's break it down step by step.

...

**4. Force Analysis at the Moment $B$ is Placed:**
- At the moment $B$ is placed, the spring has not yet had time to extend further to balance the new total weight. The spring force remains 60 N (the initial force that balances the weight of $A$).
- The net force on $A$ is the sum of the spring force and the force from $B$:$F_s + F_B = m_A g$.
- Here, $F_s = 60\,\text{N}$ and $m_A g = 60\,\text{N}$, so: $60\,\text{N} + F_B = 60\,\text{N}$.
- Solving for $F_B$, we get: $F_B = 0\,\text{N}$.

**5. Re-evaluate the Situation:**
- The above analysis is not correct because we need to consider the instantaneous force between $A$ and $B$ due to the sudden addition of $B$.
- The force between $A$ and $B$ is the force $B$ exerts on $A$, which is the weight of $B$: $F_B = m_B g = 4\,\text{kg} \cdot 10\,\text{m/s}^2 = 40\,\text{N}$.

$$\boxed{40}$$

---

🌀 **MM-EUREKA-32B**

To solve this problem, we need to determine the force between objects $A$ and $B$ at the moment $B$ is placed on top of $A$. Let's break the problem into steps:

...

**4. Applying Newton's Second Law to $A$**

$$F_{\text{net}} = F_B + F_A - F_{\text{spring}}$$
$$= F_B + 60 - 60 = F_B.$$

The net force on $A$ causes an upward acceleration $a$, which is the same as the acceleration of $B$ because they are in contact.

**5. Analyzing the Forces on $B$**
- The forces on $B$ are:
  - The weight of $B$: $F_B = 40\,\text{N}$ (downward),
  - The force from $A$: $F_{\text{contact}}$ (upward).

**6. Applying Newton's Second Law to $B$**

$$F_B - F_{\text{contact}} = m_B a.$$

**7. Finding the Acceleration $a$**
- The total mass of the system is $m_{\text{total}} = m_A + m_B = 6 + 4 = 10\,\text{kg}$.
- The net force on the system is the extra weight from $B$: $F_{\text{net, total}} = 40\,\text{N}$.
- Using Newton's Second Law for the entire system: $F_{\text{net, total}} = m_{\text{total}} a \Rightarrow 40 = 10a \Rightarrow a = 4\,\text{m/s}^2$.

**8. Finding the Contact Force $F_{\text{contact}}$**

$$F_B - F_{\text{contact}} = m_B a$$
$$40 - F_{\text{contact}} = 4 \cdot 4 = 16$$
$$F_{\text{contact}} = 40 - 16 = 24\,\text{N}.$$

<answer>The answer is $\boxed{C}$. </answer>

Figure 7: Example from physics. Red highlights Qwen2.5-VL-32B-Instruct's reasoning errors. Green shows MM-Eureka-32B's correct reasoning.

**Question:**

At room temperature, a 0.100 mol/L NaOH solution is used to titrate 25.00 mL 0.100 mol/L $H_2C_2O_4$ solution, and the resulting titration curve is shown in the figure (the volume of the mixed solution can be considered as the sum of the volumes of the solutions before mixing). Which of the following statements is incorrect? (    )

**A**. In the solution at point X: $c(Na^+) > c(HC_2O_4^-) > c(C_2O_4^{2-})$

**B**. Observing the solution at point Y indicates that the degree of ionization of $HC_2O_4^-$ is greater than the degree of hydrolysis of $C_2O_4^{2-}$

**C**. In the solution at point Z: $c(Na^+) > c(C_2O_4^{2-}) > c(HC_2O_4^-) > c(OH^-) > c(H^+)$

**D**. Throughout the titration process: $c(Na^+) + c(H_2C_2O_4) + c(HC_2O_4^-) + c(C_2O_4^{2-}) = 0.100$mol/L

**Answer:** $C$

---

🦋 **Qwen2.5-VL-32B-Instruct**

**1. Understanding the Reaction:**
- The titration is between a strong base (0.100 M NaO) and a weak dicarboxylic acid (0.100 M $H_2CO_4$).
- The reaction steps are:

  1. $H_2CO_4 + NaO \rightarrow NaHC_2O_4 + H_2O$
  2. $NaHC_2O_4 + NaO \rightarrow Na_2CO_4 + H_2O$

**2. Key Points on the pH Curve:**
- **Point X (pH ≈ 2):** This is the initial point where only $H_2CO_4$ is present. The solution is acidic.
- **Point Y (pH ≈ 5.5):** This is the first equivalence point where all $H_2CO_4$ has been neutralized to $NaHC_2O_4$ (solutions of $NaHC_2O_4$ are acidic due to the protonation of the conjugate base).
- **Point Z (pH ≈ 8.7):** This is the second equivalence point where all $NaHC_2O_4$ has been neutralized to $Na_2CO_4$ (solutions of $Na_2CO_4$ are basic due to the deprotonation of the conjugate base).

...

**4. Final Answer:** The only option that is **incorrect** is Option $\boxed{A}$.

---

🐢 **MM-EUREKA-32B**

To solve this problem, we need to analyze the titration of a weak dicarboxylic acid $H_2C_2O_4$ (oxalate) with a strong base NaOH. The titration process involves multiple steps, and we must consider the equilibrium and ionic concentrations at different points on the titration curve. Let's break the problem into parts and solve it step by step.

**1. Understanding the titration process:**

$H_2C_2O_4$ is a weak dicarboxylic acid, which undergoes two steps of dissociation:

$$H_2C_2O_4 \rightleftharpoons H^+ + HC_2O_4^-$$
$$HC_2O_4^- \rightleftharpoons H^+ + C_2O_4^{2-}$$

The titration is carried out with NaOH, a strong base, which will neutralize the $H^+$ ions step by step.

**2. Key points on the titration curve:**

**Point X:** 25 mL of NaOH added.
- At this point, the first equivalence point is reached. All $H_2C_2O_4$ is converted to $NaHC_2O_4$ (sodium hydrogen oxalate).
- The solution contains $NaHC_2O_4$, and the pH is around 2, indicating that $HC_2O_4^-$ is a weak acid.

**Point Y:** 40 mL of NaOH added.
- This is between the first and second equivalence points. The solution contains a mixture of $NaHC_2O_4$ and $Na_2C_2O_4$.
- The pH is 5.5, indicating that the solution is slightly acidic due to the dissociation of $HC_2O_4^-$ being greater than the hydrolysis of $C_2O_4^{2-}$.

**Point Z:** 50 mL of NaOH added.
- At this point, the second equivalence point is reached. All $H_2C_2O_4$ is converted to $Na_2C_2O_4$ (sodium oxalate).
- The solution contains $Na_2C_2O_4$, and the pH is 8.7, indicating that $C_2O_4^{2-}$ is a strong base.

...

<answer>The answer is $\boxed{C}$. </answer>

---

Figure 8: Example from chemistry. Red highlights Qwen2.5-VL-32B-Instruct's reasoning errors. Green shows MM-Eureka-32B's correct reasoning.

**Question:**

Scientists isolated gene A from a certain organism and dissociated it into two single strands. One of these strands was hybridized with the messenger RNA of gene A, resulting in the pattern shown in the image. An unreasonable explanation for this is (  )

**A**. Positions 1, 3, 5, and 7 are connected through complementary base pairing

**B**. Heating to a certain temperature can separate the two strands

**C**. The diagram contains 8 types of nucleotides, 5 types of bases, and 2 types of base pairs

**D**. Since fragments 2, 2, and 6 cannot bind with the messenger RNA, fragments 2, 4, and 6 belong to non-gene segments of the DNA molecule

**Answer:** $D$

---

🦕 **Qwen2.5-VL-32B-Instruct**

...

**2. Analyzing Each Option:**

...

**Option C: The figure contains 8 types of nucleotides, 5 types of bases, and 2 types of base pairings**

- In the context of RNA and DNA, there are 4 types of nucleotides in each (A, U, G, C for RNA; A, T, G, C for DNA). When they anneal, the possible base pairings are A-T (or A-U), and G-C. This means there are 4 types of nucleotides per molecule, 5 types of bases (A, T, U, G, C), and 2 types of base pairings (A-T/U, G-C).
- The statement that there are 8 types of nucleotides is unreasonable because there are only 4 types of nucleotides in each (DNA and RNA).

**Option D: Since segments 2, 4, and 6 cannot bind with the messenger RNA, segments 2, 4, and 6 belong to non-gene segments of the DNA molecule**

- This is reasonable because segments 2, 4, and 6 do not anneal with the mRNA, which suggests that these segments do not code for the protein and are likely non-coding (non-gene) segments of the DNA.

$$\boxed{C}$$

---

🐢 **MM-EUREKA-32B**

To solve this problem, we need to analyze the image and combine it with the principle of hybridization between DNA and messenger RNA (mRNA). Let's reason step by step:

...

**2. Analyzing the options:**

...

**C. The image contains 8 types of nucleotides, 5 types of bases, and 2 types of base pairings:**

- Nucleotides:
  - DNA contains 4 types of deoxyribonucleotides (dA, dT, dC, dG)
  - mRNA contains 4 types of ribonucleotides (A, U, C, G)
  - Total: 8 types of nucleotides.
- Bases:
  - DNA contains 4 bases (A, T, C, G)
  - mRNA contains 4 bases (A, U, C, G)
  - Since U replaces T in RNA, there are only 5 distinct bases (A, T/U, C, G).
- Base pairing:
  - In DNA-mRNA hybridization, the pairings are A–U, T–A, C–G, G–C.
  - However, only A–U and C–G actually form during hybridization.
  - Thus, only 2 types of base pairings exist.

**D. Since fragments 2, 4, and 6 do not hybridize with mRNA, they belong to non-genic segments of DNA:**

- The unhybridized regions (2, 4, 6) do not pair with mRNA, but this does not directly imply that they are non-genic segments.
- In genomic DNA, unpaired regions could be non-coding parts of a gene (e.g., introns, enhancers), or regulatory regions.
- Therefore, we cannot conclude that they are non-genic simply because they do not hybridize.

**3. Conclusion:**

- Option D is incorrect because we cannot determine from hybridization alone that these regions are non-genic.

<answer>The answer is $\boxed{D}$. </answer>

Figure 9: Example from biology. Red highlights Qwen2.5-VL-32B-Instruct's reasoning errors. Green shows MM-Eureka-32B's correct reasoning.

