# OpenReview forum: "MM-Eureka: Exploring the Frontiers of Multimodal Reasoning with Rule-based Reinforcement Learning"
_ICLR.cc/2026/Conference — ICLR 2026 Conference Withdrawn Submission_

### Official Review · Reviewer_A5wL · 2025-10-27

**Soundness:** 3
**Presentation:** 3
**Contribution:** 3
**Rating:** 4
**Confidence:** 4

**Summary:**

This paper addresses the issues of training instability and the lack of high-quality datasets when applying rule-based RL to MLLMs. The authors construct and introduce a new, high-quality multimodal mathematical reasoning dataset, MMK12. The training set of this dataset includes over 15600 K-12 level fill-in-the-blank questions, covering various domains such as functions and geometry, with human-verified answers and detailed CoT solutions. Experimental results show that the MM-Eureka model achieves improvements across several multimodal mathematics and multidisciplinary reasoning benchmarks, including MathVista, WeMath.

**Strengths:**

1. MMK12, as a high-quality, real-world multimodal dataset, ensures the accuracy of answers and solution processes. It covers a wide range of mathematical topics and is more reliable than other datasets relying on synthetic data or LLM-generated content. This guarantees the generalizability and accuracy of the training process.
2. The methods presented in the paper are clear, and the authors have made efforts to ensure the reproducibility of their results. The openness of the dataset, models, and code further supports the transparency of the research.

**Weaknesses:**

1. The paper explicitly states that o1 was only evaluated on a randomly selected sample of 500 instances, which weakens its comparability with other methods on the full dataset. Additionally, the result of o1 on WeMath is reported as 98.7%, which seems highly unreasonable. This raises concerns about the validity of the reported results, and I recommend that the authors carefully check their evaluation code.
2. The authors claim to have achieved SOTA on multimodal reasoning benchmarks, but some open-source SOTA models, such as InternVL3, were not compared, which might result in an overstatement of the contribution.
3. The paper does not introduce a completely new RL algorithm. The core algorithm, GRPO, is derived from DeepSeek-R1. Lack of some innovation.
4. While the model demonstrates good generalization across interdisciplinary tasks (e.g., mathematics, physics, chemistry, and biology), there is a lack of results on other domain-specific benchmarks.
5. The high quality of MMK12 depends on “human-verified CoT and answers.” While this is a strength in terms of quality, it poses scalability challenges.

**Questions:**

Same as the Weaknesses.

---

### Official Review · Reviewer_8A25 · 2025-10-27

**Soundness:** 2
**Presentation:** 2
**Contribution:** 2
**Rating:** 4
**Confidence:** 3

**Summary:**

This paper proposes a complete reinforcement learning training framework for multimodal reasoning tasks. The authors first constructed a high-quality K12 multimodal mathematical reasoning dataset, MMK12, covering fields such as function, geometry, and spatial reasoning. Then, based on Qwen2.5-VL, they introduced an online filtering mechanism and a two-stage RL training strategy, which significantly alleviated the instability and collapse problems of large-model RL training. Experiments have shown that the proposed MM-Eureka (7B and 32B) model surpasses existing open source models on multiple mathematical and interdisciplinary multimodal reasoning benchmarks.

**Strengths:**

1. It constructed a large-scale and high-quality multimodal mathematical reasoning dataset MMK12, which contains more than 15,000 samples and a multidisciplinary evaluation set, making up for the narrow coverage of existing geometry datasets (such as Geo3k).

2. Through online filtering strategies and two-stage training, the collapse problem in large-scale RL training is effectively solved. This is a practical contribution in both engineering and methodological aspects.

3. The experimental results are convincing and work well on multiple benchmarks

**Weaknesses:**

1. The novelty is limited. While the engineering implementation is excellent, the core reinforcement learning algorithm (GRPO + rule-based reward) is largely inherited from DeepSeek-R1, lacking any theoretical or algorithmic innovation. Besides, Although online filtering and two-stage training are effective, they seem "empirical engineering tricks" rather than novel training principles.

2.. The models compared in the paper are somewhat outdated. It is recommended to add the latest models such as GPT-5 and Gemini2.5pro for comparison. Besides, The open source model is also a bit outdated. It is recommended to use the latest SOTA open source model such as InternVL3.5[1].

[1] Wang W, Gao Z, Gu L, et al. Internvl3. 5: Advancing open-source multimodal models in versatility, reasoning, and efficiency[J]. arXiv preprint arXiv:2508.18265, 2025.

3. Potential overfitting to internal benchmarks: There is a risk that MMK12, as both a training and evaluation source, could introduce bias if not carefully separated (the paper attempts to clarify but should make this completely explicit and consider reporting alternative, wholly external benchmarks or variants).  I think we can test more difficult questions or other reasoning, non-knowledge-based multimodal benchmarks, such as VisualPuzzles[2] and VisuLogic[3].

[2]. Song Y, Ou T, Kong Y, et al. VisualPuzzles: Decoupling Multimodal Reasoning Evaluation from Domain Knowledge[J]. arXiv preprint arXiv:2504.10342, 2025.

[3]. Xu W, Wang J, Wang W, et al. Visulogic: A benchmark for evaluating visual reasoning in multi-modal large language models[J]. arXiv preprint arXiv:2504.15279, 2025.

**Questions:**

Seeing weaknesses

---

### Official Review · Reviewer_nZrb · 2025-11-03

**Soundness:** 3
**Presentation:** 4
**Contribution:** 2
**Rating:** 4
**Confidence:** 3

**Summary:**

This paper introduces a training framework with rule-based RL for VLMs, significantly enhancing their complex reasoning abilities. It proposes a new large mathematic dataset MMK12, along with a cross-subject evaluation dataset, which may be beneficial to the VLM community. This paper trained 7B and 32B models on the proposed MMK12 datasets, and introduce a novel two-stage training strategy to improve the performance of the large model. Experiments demonstrate that the trained model can achieve comparable ability in task on various subjects.

**Strengths:**

1. this paper proposes a new dataset MMK12, which contains a large number of tasks across various math domains.
2. this paper brings some insight on the training and fine-tuning of MultiModal LLM models, and explains how the whole framework works in detail. The representation of this paper is good and clear. Experiments illustrate that the trained models can achieve comparable performance compared to large LLM models.

**Weaknesses:**

1. The whole training process are combination of existing training techniques. Only a two-stage training strategy utilized in training stage of the 32B model.
2. The MMK-12 dataset lacks multi-language samples, which may limits the performance of model.

**Questions:**

1. It is said in Sec 4.4 that training the 32B model only on MMK12 shows performance degradation in domains like geometry. However MMK12 consists mainly (60%) of task about geometry. And combing another dataset(Geo3K) enhances training stability. What is causing this?
2. Most of the experiments are carried out on the proposed MMK12 dataset. Can authors extend experiments on other (e.g. smaller) dataset to show the robustness of proposed training framework?

---

### Note · Authors · 2025-11-24

I have read and agree with the venue's withdrawal policy on behalf of myself and my co-authors.